# The Chloroplast of *Chlamydomonas reinhardtii* as a Testbed for Engineering Nitrogen Fixation into Plants

**DOI:** 10.3390/ijms22168806

**Published:** 2021-08-16

**Authors:** Marco Larrea-Álvarez, Saul Purton

**Affiliations:** 1School of Biological Sciences and Engineering, Yachay-Tech University Hacienda San José, Urcuquí-Imbabura 100650, Ecuador; malarrea@yachaytech.edu.ec; 2Algal Research Group, Department of Structural and Molecular Biology, University College London, Gower Street, London WC1E 6BT, UK

**Keywords:** nitrogenase, synthetic biology, transplastomic engineering, *Chlamydomonas reinhardtii*, NifV

## Abstract

Eukaryotic organisms such as plants are unable to utilise nitrogen gas (N_2_) directly as a source of this essential element and are dependent either on its biological conversion to ammonium by diazotrophic prokaryotes, or its supply as chemically synthesised nitrate fertiliser. The idea of genetically engineering crops with the capacity to fix N_2_ by introduction of the bacterial nitrogenase enzyme has long been discussed. However, the expression of an active nitrogenase must overcome several major challenges: the coordinated expression of multiple genes to assemble an enzyme complex containing several different metal cluster co-factors; the supply of sufficient ATP and reductant to the enzyme; the enzyme’s sensitivity to oxygen; and the intracellular accumulation of ammonium. The chloroplast of plant cells represents an attractive location for nitrogenase expression, but engineering the organelle’s genome is not yet feasible in most crop species. However, the unicellular green alga *Chlamydomonas reinhardtii* represents a simple model for photosynthetic eukaryotes with a genetically tractable chloroplast. In this review, we discuss the main advantages, and limitations, of this microalga as a testbed for producing such a complex multi-subunit enzyme. Furthermore, we suggest that a minimal set of six transgenes are necessary for chloroplast-localised synthesis of an ‘Fe-only’ nitrogenase, and from this set we demonstrate the stable expression and accumulation of the homocitrate synthase, NifV, under aerobic conditions. Arguably, further studies in *C. reinhardtii* aimed at testing expression and function of the full gene set would provide the groundwork for a concerted future effort to create nitrogen-fixing crops.

## 1. Introduction

Diazotrophs are bacterial and archaeal species that are capable of converting N_2_ into NH_3_ in a process known as biological nitrogen fixation (BNF) that is catalysed by the metallocluster enzyme nitrogenase [1,2]. Three different classes of nitrogenase have been described to date, with each consisting of two metalloenzyme components—an ATP-dependent reductase referred to as the Fe protein, and its catalytic partner the dinitrogenase, or FeX-co protein. The three nitrogenase classes are distinguished by the metal species within the active site cofactor where X is either molybdenum (Mo), vanadium (V), or iron (Fe), but share a related structure [3]. The Fe protein is a homodimer with the apoprotein encoded by the *nifH* gene in the Mo class, and by the *vnfH* and *anfH* genes for the V and ‘Fe-only’ versions, respectively. The FeX-co protein is a heterotetramer composed of two α and two β subunits encoded by *nifD* and *nifK* for the Mo version, and *vnfD* and *vnfK* for the V version. The dinitrogenase of the alternative Fe version has an extra subunit encoded by *anfG* in addition to the AnfD and AnfK subunits. The catalytic cofactor, a [7Fe-9S-C-X-homocitrate] cluster, is assembled independently of the structural subunits and is inserted only later into the complex to generate the active form [3,4]. Ammonia production involves the transfer of eight electrons to the Fe protein, which in turn, provides these electrons to the dinitrogenase protein for N_2_ reduction, with the concomitant hydrolysis of two ATP molecules per electron [5]. The biosynthesis of the Mo nitrogenase is summarised in Figure 1.

Fixed forms of nitrogen play a critical role in global food production. However, the contribution made by BNF to crop production falls far short of the requirements of modern agriculture. Consequently, chemical fertilisers are used extensively to meet the nitrogen needs of intensively farmed crops and animal pasture [6]. Such fertiliser production is based on the Haber–Bosch chemical process of converting N_2_ and H_2_ into ammonia, and whilst this process has been a key driver in the Green Revolution there is increasing recognition of its environmental consequences, economic costs, and unsustainability [7,8]. These issues include the use of ~2% of the total fossil fuel supply during fertiliser production and the associated release of vast volumes of CO_2_ into the atmosphere, the release of significant amounts of nitrous oxides during application of the fertilisers onto the field, and the eutrophication of aquatic ecosystems as a consequence of nitrate run-off.

To reduce our dependency on industrial fertilisers, several biological solutions have be proposed that aim to engineer N_2_ fixation into crops, either indirectly by establishing root symbiosis with diazotrophic bacteria similar to that seen in leguminous plants, by transferring nitrogenase to bacteria that associate naturally with cereal crops, or directly by introducing the nitrogenase enzyme into the plant itself [9,10,11,12,13,14]. In a pivotal study relating to the last strategy, the nitrogen fixation (*nif*) genes from *Klebsiella pneuminiae* were successfully expressed in *E. coli*, allowing this non-diazotrophic bacterium to survive in the absence of a source of fixed nitrogen [15]. *E. coli* has proved to be a valuable host for engineering diazotrophy, as many studies have used it for determining the minimal transgene requirements for N_2_ fixation [16], or redesigning *nif* clusters for engineering purposes [17,18]. These results have encouraged efforts to incorporate the trait into eukaryotic organisms, and several groups have reported the production of nitrogenase components in yeast and plant cells with a focus on targeting the Nif proteins into the mitochondrion [19,20]. This organelle was chosen both because it is a major site of ATP synthesis, and a minimum of 16 ATP molecules are needed to fix one molecule of N_2_ (Figure 1), and because the oxygen consumption in the mitochondrion should protect the highly oxygen-sensitive nitrogenase from inactivation [21]. However, these studies have highlighted some of the challenges of targeting Nif proteins into the mitochondrion, including undesired N-termini modifications and instability of the proteins as a result of processing by mitochondrial processing peptidases [22,23]. For instance, the initial study of Nif proteins targeted to the yeast mitochondrion identified undesirable processing of the core subunit NifD [24].

The chloroplast has also been proposed as a potential location for a eukaryotic nitrogenase [25,26,27,28], as this energy-conversion organelle generates the high levels of ATP necessary and possesses the electron transfer systems (ferredoxin–NADPH oxidoreductases and ferredoxins) that can supply the reductant required by the Fe protein [26]. Moreover, chloroplast genetic engineering is feasible for several species, and *nif* genes integrated into chloroplast genomes have been shown to give rise to functional products [27,28]. Direct expression within the organelle circumvents the need for import of the Nif proteins from the cytosol and also offers the potential of much higher expression levels [29]. However, two major challenges must still be overcome. First, nitrogenase’s extreme sensitivity and irreversible inactivation by molecular oxygen conflicts with the primary activity of chloroplast—oxygenic photosynthesis [25]. Secondly, the assembly of the active holoprotein is very complex, with at least nine genes being necessary for expression in bacteria [16]. Thus, nitrogenase expression needs to be temporally (e.g., during the dark phase of a diurnal cycle) or spatially (e.g., in non-green plastid types) separated from photosynthesis, and the synthesis of each Nif component needs to be optimised and correctly regulated to allow efficient assembly of the complex. Clearly, such engineering of the chloroplast requires a simple, genetically tractable laboratory system that can serve as an initial testbed for the many design iterations necessary to achieve this goal. Single-cell green algae seem well-suited for such research, as they are easy to culture in the lab with short generation times. Furthermore, some species are capable of growth in the dark when supplied a fixed carbon source or are capable of phototrophic growth under anaerobic conditions, both of which would permit the testing of nitrogenase assembly and function under conditions of reduced oxygen. Furthermore, the facultative heterotrophy can be exploited to isolate photosystem II mutants that are incapable of water splitting, allowing studies in the light and under aerobic conditions, but with a low oxygen potential in the chloroplast. Optimisation of plastid nitrogenase biogenesis could be pursued in these platforms, and the technology then transferred to plant models and ultimately to key crop species such as rice.

The green single-cell alga *Chlamydomonas reinhardtii* is a widely used model for eukaryotic cell biology and has been referred to as the “green yeast” owing to its simple and rapid cultivation, well-characterised genetics, extensive DNA transformation tools, and detailed ‘omics data [30,31,32,33]. Chloroplast transformation is well established in *C. reinhardtii*, allowing the targeted insertion of transgenes into the organelle’s small genome (=plastome) and their stable, high-level expression [34]. The last few years has seen a growing interest in using genetic engineering approaches to introduce a novel metabolism in the algal chloroplast: building on the native biosynthetic pathways within the organelle [35] and exploiting the niche of this sub-cellular compartment [36]. With regard to engineering nitrogen fixation, a key observation is that *C. reinhardtii* cells are able to grow in the absence of oxygen [37]. Under these conditions, several redox enzymes specifically accumulate in the chloroplast [38]; an illustration of what could be feasible with nitrogenase. In a pioneering study, the chloroplast synthesis of the NifH protein was demonstrated in *C. reinhardtii* [27]. Interestingly, the *nifH* gene was able to complement a mutation in a similar chloroplast gene (*chlL*) which encodes a subunit of an endogenous enzyme (the light-independent protochlorophyllide reductase) that is related to nitrogenase. This complementation confirmed that the Fe protein was functional and able to couple ATP hydrolysis to electron transfer, and provides an encouraging starting point for nitrogenase research using this model system. In addition, the successful engineering of an Fe-only nitrogenase in bacteria [16], along with the expression of Fe-only nitrogenase proteins in the mitochondrial matrix in yeast [39], indicates that a focus on this nitrogenase class would bypass the issue of limited molybdenum availability in the chloroplast.

In this mini-review, we discuss the accessory endogenous components within the chloroplast that could support an active nitrogenase, and propose a minimal set of transgenes that could be used as a foundation for creating a diazotrophic *C. reinhardtii*. As a further proof-of-concept building on the NifH work, we show that the homocitrate synthase NifV (Figure 1) can be expressed from the plastome and accumulated under aerobic conditions.

## 2. Nitrogenase in the Chloroplast

As noted above, the chloroplast has long been considered as a potential sub-cellular location for nitrogenase [25]. Importantly, this photosynthetic organelle harbours electron transfer complexes and electron carriers that contain various classes of iron-sulphur (Fe-S) clusters. As the chloroplast contains the basic machinery for synthesis of Fe-S clusters and attachment to endogenous apoproteins, it is possible that this machinery could also serve for nitrogenase biosynthesis (Figure 1). Indeed, the production in the *C. reinhardtii* chloroplast of a functional NifH containing a 4Fe-4S cluster illustrates this potential [27]. It has been proposed that reducing power derived from photosynthesis during the day could be coupled to the Fe protein via endogenous ferredoxins during the night, with synthesis of the nitrogenase controlled on a diurnal cycle [26]. Various methods for inducible expression of chloroplast transgenes have been developed and could be adapted to allow the regulation of key *nif* genes in the plastome under a circadian or dark/light cycle [40,41]. Alternatively, accumulation of Nif proteins in other plastid types present in the non-green tissue of plants has been proposed as a strategy to circumvent the issue of oxygen sensitivity. This approach is complicated by the much lower levels of endogenous gene expression in these plastids, although transgene design could mitigate this to improve expression levels of the *nif* genes [42]. However, the biggest technical challenges to engineering plant plastids are: (i) The current lack of robust chloroplast transformation methods for most crop species, meaning that initial studies would probably have to be conducted using *Nicotiana tabacum* (i.e., tobacco) where chloroplast engineering is most advanced [43]; (ii) The extended time required to create a stable chloroplast transformant when working with higher plants (typically several months) and the large greenhouse space required to generate multiple different transgenic plants in parallel. This severely limits the speed and breadth of the ‘design-build-test-learn’ cycle that is central to an iterative research strategy where multiple parameters are explored to optimise the design, expression, and regulation of a set of transgenes [44]; (iii) The multicellular nature and obligate phototrophy of plants that inevitably complicates the assembly and testing of nitrogenase designs *in planta*.

## 3. The *Chlamydomonas* Chloroplast as a Testbed

*C. reinhardtii* cells possess a single chloroplast that occupies ~40% of the cell volume [31]. Like its higher plant counterpart, the algal chloroplast contains a prokaryote-derived genetic system that has evolved from the original cyanobacterial ancestor [45]. The small plastome houses ~100 genes that mainly encode core components of the photosynthetic complexes or the organelle’s transcription–translation machinery [46]. DNA transformation of the *C. reinhardtii* chloroplast was first demonstrated in 1988, and since that time an extensive set of molecular tools has been developed to allow routine plastome engineering, including the introduction and regulated expression of transgenes [47]. Although most recombinant proteins synthesised to date have been therapeutic or general proteins [34,36], progress has also been made in the expression of enzymes needed for synthesis of novel metabolites within the chloroplast [48,49,50,51,52]. Hence, the algal chloroplast is increasingly being recognised as a potential site for light-driven metabolic engineering [36]. Recent studies reporting the expression of multiple transgenes from the plastomes of both *C. reinhardtii* [53,54] and tobacco [43] demonstrate that complex genetic engineering, such as that required for the introduction of multi-step metabolic pathways or multi-subunit enzyme complexes, is feasible within the chloroplast. Currently, the main technical challenges for the successful expression of a set of *nif* genes in the *C. reinhardtii* chloroplast relate to the need for advanced synthetic biology (SynBio) tools: namely, (i) a need for a standardised DNA assembly method that would allow a rapid SynBio approach to the building and testing of multiple transgene configurations, and (ii) an increased number of validated regulatory DNA parts and inducible mechanisms for tuneable regulation of these transgenes. This is an area of active research [55,56,57,58] and it is likely that we will see significant advances in algal chloroplast SynBio in the next few years.

The chloroplast of *C. reinhardtii* harbours several reductase enzymes which share common features with nitrogenase, are O_2_ sensitive, and accumulate under dark anaerobic conditions. These enzymes include the [FeFe]-hydrogenases (HYDA) and the dark-operative protochlorophyllide oxidoreductase (DPOR) [17,59], and are known to obtain electrons from various ferredoxin isoforms (e.g., FDX1), some of which are activated during anaerobiosis [17,60]. As previously mentioned, expression of *nifH* into the *C. reinhardtii* chloroplast was able to complement a DPOR mutant lacking *chlL*, allowing chlorophyll biosynthesis in the dark. ChlL is a [4Fe-4S] subunit of DPOR that is closely related to NifH (see below), and the complementation demonstrated that the metal clusters of the Fe protein were supplied and properly assembled into the NifH apoprotein using the endogenous mechanisms in the chloroplast, and that electron and ATP supply were coupled to the hybrid enzyme [27]. N_2_ fixation is a costly process requiring significant amounts of ATP and reducing power, and in the chloroplast, this would be derived principally from photosynthetic processes. Nevertheless, an active nitrogenase might represent a heavy burden for the physiology of the cell, with energy and reductant diverted away from carbon fixation, resulting in sickly, slow-growing transgenic lines. However, *C. reinhardtii* is capable of mixotrophic growth by taking up and metabolising (via the glyoxylate pathway) exogenous acetate as a reduced form of carbon, thereby supplementing the photosynthetic generation of ATP and reducing power. Additionally, this green alga can thrive in anaerobic conditions due to fermentation circuits, allowing ATP synthesis in the absence of oxygen [37], which is particularly attractive for the study of oxygen-sensitive metallocluster enzymes. Interestingly, ATP and reductant have been observed to be imported from mitochondria to chloroplasts with the purpose of setting up proper conditions for photosynthesis before the onset of light [37]. This metabolic flexibility of *C. reinhardtii* bodes well for the tolerance and support of an active nitrogenase within the chloroplast.

## 4. A Minimal Set of Nitrogenase Genes

The assembly pathway for nitrogenase is complex and requires the coordinated expression and interaction of multiple subunits (Figure 1). Heterologous expression of *nif* genes in bacterial systems has highlighted the importance of reducing the genetic requirements by eliminating non-essential genes and regulatory elements. Pioneering studies using *E. coli* have demonstrated that reorganisation and refactoring of *nif* clusters allowed their successful expression [16,17,23]. In particular, the engineering of a 10-gene Fe-only nitrogenase provides a conceptual blueprint for bringing nitrogen fixation into the chloroplast, as expression of a molybdenum-containing nitrogenase would be difficult owing to the scarcity of molybdenum in the organelle. This minimal system combined the *Klebsiella oxytoca nifBUSV* genes needed for cluster assembly and the *nifF* and *nifJ* genes needed for electron supply, with the structural genes *anfHDGK* from *Azotobacter vinelandii* (Figure 2A) [16]. The AnfHDGK proteins have recently been shown to accumulate in transgenic yeast when targeted to the mitochondrial matrix, with a stable apo-AnfDK forming [39]. From this set of ten genes, at least two could be considered non-essential for nitrogenase activity in the chloroplast, as a study in *E. coli* has shown that plastid ferredoxins and ferredoxin-NADPH oxidoreductases (FNRs) are able to partially complement the NifJ–NifF module as the electron donor to both MoFe- and Fe-only nitrogenases [26]. This indicates that the native plastid machinery could directly supply reducing power for chloroplast nitrogen fixation. Green algae possess a wide variety of ferredoxins with ferredoxin 1 (FDX1) being the most promiscuous, as it transfers electrons from photosystem I to a number of proteins [59]. FDX1 interacts principally with ferredoxin-NADP+ oxidoreductase (FNR) for NADP+ reduction [60] but, along with other ferredoxin isoforms (activated during anaerobiosis), FDX1 is known to donate electrons to other redox proteins including the [FeFe]-hydrogenase HYDA [61]. It is therefore possible that the genes for NifJ and NifF could be excluded from the minimal set of chloroplast transgenes with FDX1 serving as the reductant donor.

To further reduce the number of nitrogenase genes, the NifU/NifS system responsible for metal cluster biosynthesis (Figure 1) might also be replaced by endogenous modules. When *nifH* was expressed in the *C. reinhardtii* or tobacco chloroplast, the [4Fe-4S] cluster was successfully assembled into the apoprotein using endogenous components as substitutes for the NifU/NifS system [27,28]. However, it should be noted that the NifH activity in the plant chloroplast was low, suggesting that NifU and NifS expression might be needed for full activity in this case [28]. Indeed, a separate study where Nif proteins were synthesised in the cytosol and targeted into the tobacco chloroplast found that, NifH activity required co-expression of NifU and NifS [63]. Green algae and higher plants are known to encode a number of iron-sulphur cluster assembly proteins and cysteine desulfurases [64]. Our in silico analysis of the *C. reinhardtii* nuclear genome reveals the presence of genes for NifU/NifS-like enzymes (XP_001695148.1 and XP_001696892.1, respectively. See Appendix A). Both enzymes are likely to be located in the chloroplast [65], although no transit peptides could be identified. These endogenous enzymes could potentially provide the components needed for the biosynthesis of nitrogenase [4Fe-4S] clusters and eliminate the requirement for *nifU* and *nifS* within the gene set. Thus, we suggest that a minimal set of six genes, encoding the *A. vinelandii* structural subunits (*anfHDGK*) and the *K. pneumoniae* biosynthetic components (*nifBV*), could be used as a basis for investigating nitrogen fixation in eukaryotic algae (Figure 2B,C); although, including the NifU/NifS system might prove crucial for securing full activity of the enzyme. The expression of multiple transgenes in the chloroplast of *C. reinhardtii* is certainly achievable, and the six genes could be expressed either as separate transcriptional units [54] or as several small operons [53] with gene cluster inserts at several neutral loci within the plastome [65]. However, it will be critical to tune the strength of expression of each gene, with the aim of achieving the correct stoichiometric levels of each *anf/nif* gene product.

## 5. Expression of the *anf*/*nif* Genes in the Chloroplast of *C. reinhardtii*

As discussed above, DPOR is a chloroplast enzyme whose sequence is similar to that of nitrogenase (Figure 3A). It is involved in the production of chlorophyll in the dark, a pathway present in algae and gymnosperms but absent in angiosperms [66]. The DPOR complex is encoded by a triad of chloroplast genes (*chlL*, *chlB*, and *chlN*) with the primary sequences of ChlL, ChlN, and ChlB being highly similar to nitrogenase’s NifH, NifD, and NifK, respectively [64]. The *K. pneumoniae nifH* sequence was able to rescue a Δ*chlL* mutant to a wild-type ‘green-in-the-dark’ phenotype, confirming that the supply and assembly of the iron-sulphur cluster into the NifH enzyme can be achieved using the endogenous system (rather than NifU and NifS) and that the enzyme is sufficiently active to support protochlorophyllide reduction by interfacing with endogenous electron donors (e.g., ferredoxins, FNR). Alignment of the protein sequences reveal that NifH, AnfH, and ChlL possess consensus regions serving as Fe-S cluster ligands and nucleotide-binding motifs (Figure 3B). Our in silico analysis shows that both share a similar level of similarity to ChlL:AnfH (37%) and NifH (34%) (see Appendix A). Arguably, therefore, accumulation of functional AnfH in the plastome appears attainable and should be of primary focus.

Currently, there are no examples of the other nitrogenase components expressed in *C. reinhardtii*, although their accumulation has been achieved in plant and yeast mitochondria [19,20,21,24,63,67]. Again, plastid accumulation of the Anf structural subunits (Figure 2C) should be achievable with the correct choice of regulatory elements and environmental conditions (e.g., anaerobiosis) to optimise protein expression. The (SAM)-dependent NifB is required for the addition of the carbon atom to the Fe-S metal structure to generate the so-called NifB-co cluster (Figure 1). Assembly of the chloroplast-localised hydrogenase HYDA1 also relies on radical SAM chemistry, and the (SAM)-dependent maturases HYDG and HYDE are likely to be chloroplast-localised as well [61]. Co-expression in yeast of the nitrogenase maturation proteins NifU, NifS, and FdxN from *Azotobacter vinelandii* with NifB from the archaea *Methanocaldococcus infernus* or *Methanothermobacter* yielded a functional NifB protein within the mitochondrial matrix [20]. Likewise, transient expression of NifB in *Nicotiana benthamiana* resulted in accumulation in the plant’s mitochondrial matrix [19]. NifB expression should be feasible in the chloroplast and the requirement (or not) of other genetic determinants such as NifU and NifS testable, especially under dark, anaerobic conditions.

NifV represents another key component of the proposed nitrogenase system, as it is responsible for generating the organic element of the primary cofactor. NifV is a homocitrate synthase whose product (homocitrate: (R)-2-hydroxy-1,2,4-butanetricarboxylic acid) is coupled to the catalytic cluster (Figure 1, Figure 2C). This organic moiety is known to contribute to the structural assembly and overall redox properties of the cofactor [68]. NifV has been shown to accumulate in the plant mitochondrial matrix and yeast [19,69]. This enzyme catalyses the condensation of acetyl coenzyme A and 2-oxoglutarate to form homocitrate and CoA. Thus, as a proof-of-concept experiment, we decided to introduce a codon-optimised version of *nifV* from *K. oxytoca* into the plastome to determine if such an enzyme could be synthesised in the chloroplast, and if its expression would affect microalgal growth rates. The coding sequence, which also encoded the haemagglutinin epitope tag (HA-tag), YPYDVPDYA at the C terminus [70], was assembled into the pSRSapI transformation vector [71]. This vector carries a functional version of the endogenous *psbH* gene and thus allows selection for transformation lines by means of phototrophic restoration of a Δ*psbH* strain, TN72 (Figure 4A). Transformation was via the glass-bead vortexing method [72] with minimal medium used to select directly for transformant colonies [73].

As shown in Figure 4B, PCR analysis revealed that the selected phototrophic colonies contained the *nifV* transgene, and after being restreaked at least twice the polyploid plastome was homoplasmic for the engineered change. SDS polyacrylamide gel electrophoresis was used to fractionate the cell lysates, which were then probed using an antibody against the HA epitope. Western blot analysis confirmed that the four colonies analysed produced the recombinant enzyme (Figure 4C), although levels were markedly lower than that seen for a control transformant engineered to make an HA-tagged serine protease (SplB) [54]. A transformed line lacking any transgenic coding sequence was used as a negative control. Expression of the homocitrate synthase during the exponential and stationary phases of growth did not have any obvious negative effect on aerobic cultures, as the crNifV3 cell line grew at a similar rate to the crSplB and TN-E controls (Figure 4D). The endogenous *psaA-1* promoter/5′UTR element used to drive *nifV* expression is considered a strong promoter [70], thus the low levels of NifV probably reflect a high rate of protein turnover. However, accumulation of the enzyme did not change during growth under anaerobic conditions (data not shown), demonstrating that NifV accumulation appears not to be oxygen sensitive, despite an earlier report that the activity of recombinant NifV purified from *E. coli* was sensitive to oxygen [68]. Indeed, accumulation of *K. oxytoca* NifV has been demonstrated in *N. benthamiana* and *S. cerevisiae* under normal aerobic conditions [19,20]. Nevertheless, anoxic induction of *nifV* gene expression in the chloroplast could improve yield, and would be feasible using a system developed for nuclear control of chloroplast transgenes [74] linked to an anoxic-specific nuclear promoter [75]. Alternatively, using *nifV* genes from other diazotroph species might identify those more resistant to proteolysis in the chloroplast.

The need to solve such problematic intermediary stages of nitrogenase biosynthesis highlights the requirement for a simple, genetically tractable testbed to explore this complex design space. Once sufficient synthesis and stability of the multiple components has been achieved, the next challenge in this endeavour is demonstrating the functionality of each protein.

## 6. Conclusions

As global food supply depends heavily on industrial fertilisers, there has been a longstanding interest in exploring biotechnological alternatives, such as the engineering of crop species to directly access N_2_. Despite encouraging preliminary results, the use of plant organelles as compartments for housing an active nitrogenase has still to overcome major challenges. The enzyme is highly sensitive to oxygen, and its biosynthesis depends on the coordinated expression of multiple genes and assembly of the complex with various elaborate cofactors. Nitrogenase accumulation at night or in non-green tissues has been suggested as a potential strategy to circumvent the O_2_ issue. Moreover, expression of a minimal set of genes from the plastome could reduce the genetic complexity and also avoid the use of transit peptides for organellar import. However, plastome engineering in plants is slow, and complicated by their obligate phototrophy and multicellular nature. This severely limits the high-throughput SynBio strategies that would be required to explore the complex design space that underlies successful nitrogenase biosynthesis. The use of a simpler and more flexible photosynthetic platform as a SynBio testbed could accelerate this effort, allowing multiple studies in parallel and more rapid circuits of the ‘design-build-test-learn’ cycle. *C. reinhardtii* represents a suitable option as it provides the advantages of rapid microbial cultivation, flexible metabolism, including anaerobic and heterotrophic growth, and well-established molecular tools for plastome engineering. Moreover, its chloroplast harbours various enzymes that have similar properties to nitrogenase and whose related machinery could be diverted to sustain N_2_ fixation. In an attempt to reduce the complexity of engineering, we propose a minimal set of six genes for expression of a plastid-encoded Fe-only nitrogenase. Multigenic engineering of the *C. reinhardtii* plastome is possible, and the next few years will see further advances in the molecular techniques needed for producing complex multi-subunit enzymes. Nevertheless, detailed optimisation of intermediary steps will be critical before coupling the complete pathway to chloroplast metabolism. As a first step, we have shown that the homocitrate synthase, NifV, can accumulate stably in the *C. reinhardtii* chloroplast, albeit at a low level, and we are confident that this green microalga can contribute significantly to the long-term goal of generating diazotrophic crops.

## Figures and Tables

**Figure 1 ijms-22-08806-f001:**
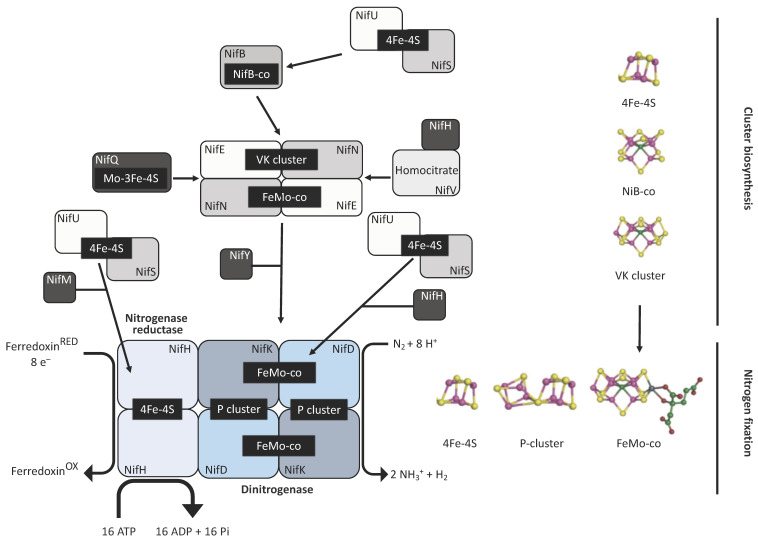
Mo nitrogenase biosynthesis. The enzyme is made of two components, the dinitrogenase reductase (Fe protein) and the dinitrogenase itself (MoFe protein). All the catalytic cofactors contain Fe-S clusters, which are provided by the NifU/NifS system. Sulphur (S) atoms are taken from L-cysteine by the action of NifS. These atoms are assembled into clusters along with Fe ions in the NifU scaffold protein. These Fe-S clusters are then transferred to the apoproteins of the complex. The Fe protein (NifH) acquires these clusters following the action of a maturation factor (NifM). The same system provides the clusters upon which the catalytic cofactors of the dinitrogenase element will be assembled. A functional MoFe protein (NifDK) requires the synthesis and assembly of the P and the FeMo-co. The assemblage of the P cluster relies on the reduction of two 4Fe-4S clusters by NifH into a single 8Fe-7S cluster. On the other hand, the FeMo-co cofactor is assembled outside the NifDK component. In the early steps, 4Fe-4S clusters are transferred to the NifB protein, which is an S-adenosylmethionine (SAM)-dependent enzyme that inserts the C atom into the cluster to form the NifB-co. This newly formed structure is then passed (via NifX or directly) to another scaffold protein made by the products of *nifE* and *nifN*. Here, the NifB-co is transformed into a structure known as the VK cluster (8Fe-9S-C). The NifEN protein contains structural 4Fe-4S clusters, and also a Mo-3Fe-4S cluster. The latter is formed in NifQ, where a molybdenum atom is assembled with an Fe-S cluster provided by the NifU/NifS system, and then transferred to NifEN. Upon interaction between NifH and NifEN, the Mo atom and the homocitrate molecule (provided by NifV) are incorporated into the VK cluster, giving rise to the FeMo-co cofactor, which is then transferred by a chaperone (NifY), or directly, to the apo-NifDK to generate the final holoenzyme. Yellow: S atoms; purple: Fe atoms; blue: molybdenum atom; light green: carbon atom; green: homocitrate.

**Figure 2 ijms-22-08806-f002:**
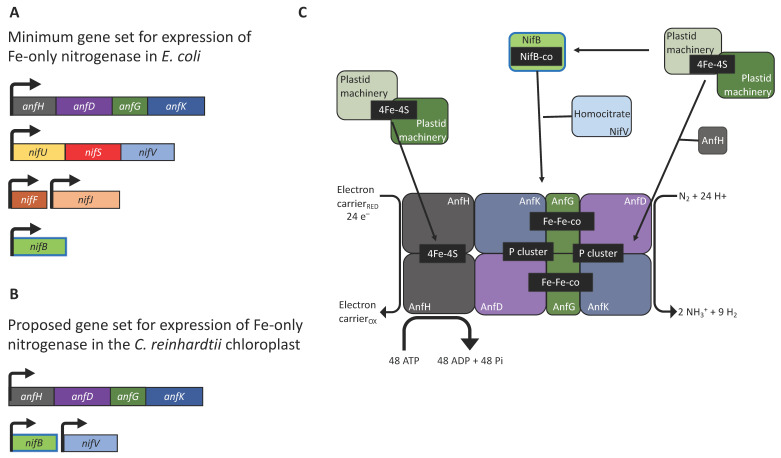
Prospects for nitrogenase engineering in the chloroplast of *C. reinhardtii*. (**A**) Minimal cluster of accessory/structural genes required for nitrogenase activity in *E. coli* [15]. (**B**) Suggested minimal set of genes required for a chloroplast nitrogenase. (**C**) Schematic representation of the hypothetic iron-only nitrogenase. The endogenous chloroplast system for assembling Fe-S clusters could provide the catalytic cofactors for cluster assembly. The (SAM)-dependent NifB is required for the fusion of two [4Fe-4S], the addition of a carbon atom, and an additional sulphur to produce the NifB-co [8Fe-9S-C], while NifV provides the homocitrate molecule. The Fe-only nitrogenase is less efficient than its Mo counterpart as it requires as many as 48 ATP molecules and 24 low-potential electrons to reduce one N_2_ molecule [62].

**Figure 3 ijms-22-08806-f003:**
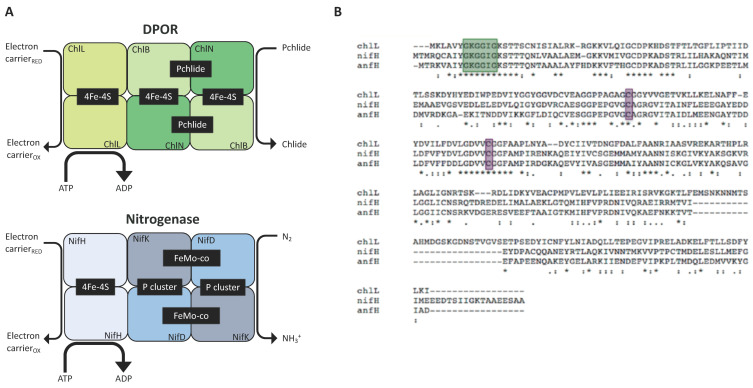
Subunit comparison of DPOR and nitrogenase. (**A**) Chlorophyll synthesis in the dark involves the reduction of the precursor molecule protochlorophyllide (Pchlide) to chlorophyllide (Chlide) by the nitrogenase-like DPOR (composed of subunits ChlL, ChlN, and ChlB) (upper panel). The nitrogenase complex is made of the homologous subunits NifH, NifD, and NifK and reduces N_2_ to NH_3_^+^ (Lower panel). Both enzymes rely on protein–protein interaction between their sub-complexes. The heterotetrametric proteins (ChlN/ChlB, NifD/NifK) interact with the dimeric component (ChlL, NifH) in an ATP-dependent fashion. Redox-active metal clusters allow electron transfer to the substrates to generate the final product (Chlide, NH_3_^+^). (**B**) Protein sequence alignments of the reductases ChlL from *C. reinhardtii*, NifH from *K. oxytoca* and AnfH from *A. vinelandii*. The three enzymes display the conserved cysteine groups (purple) that are required as ligands for the [4Fe-4S] clusters; the motif responsible for nucleotide binding is shown in green.

**Figure 4 ijms-22-08806-f004:**
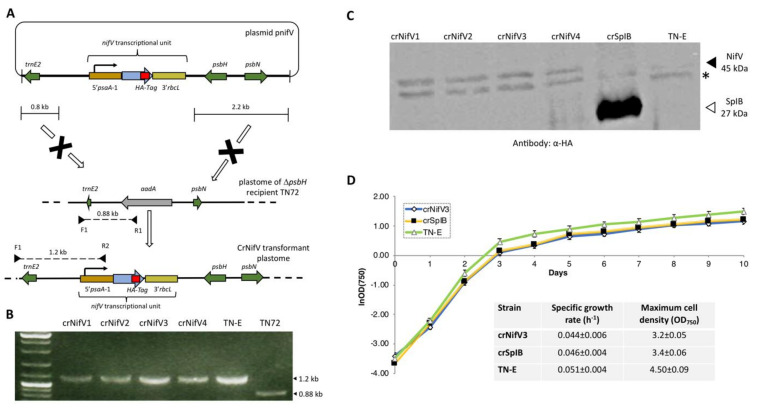
Cloning and expression of *nifV* from the plastome of *C. reinhardtii*. (**A**) Insertion of the coding sequence into the chloroplast genome was performed using a methodology described previously [71,72]. The transcriptional unit consisted of the *psaA*-1 promoter/5′ UTR, the *nifV* gene, and the *rbcL* 3′ UTR. Both the cassette and the functional version of the *psbH* gene were flanked by sequences needed for homologous recombination, which allowed rescue of the Δ*psbH* recipient strain TN72. In this strain the *psbH* gene has been replaced with an *aadA* antibiotic-resistance cassette. After transformation, the expression cassette is integrated into an intergenic locus downstream of *psbH*. (**B**) Colony PCR and agarose gel analysis confirmed the incorporation of the *nifV* transcriptional unit in four transformant lines. The strain TN72 was used as a negative control, while a strain (TN-E) generated using an “empty” vector was utilised as a positive control. The F1 primer is targeted to the 5′ region upstream of the homologous recombination site, whereas the reverse primers are targeted either to the *aadA* cassette or the *nifV* transcriptional unit. The 1.2 kb band confirms the insertion of the expression cassette gene into the plastome, while the 0.88 kb band corresponds to the plastome of TN72. The absence of such a band in the transgenic lines indicates that the polyploid plastome is homoplasmic. (**C**) Western blot analysis shows the accumulation of NifV. Cell extracts of four transformant lines (crNifV1-4) were analysed using α-HA epitope tag antibodies as described [70]. Negative (TN-E) and positive (crSpIB) control transformants were included in the analysis. As a loading control, antibodies against the endogenous RbcL were used. (*) denotes the presence of an unknown endogenous protein detected by the α-HA antibodies. (**D**) Expression of recombinant NifV did not have major effects on culture growth. The graph shows growth curves for the crNifV3, crSpIB, and TN-E transformants. For calculating specific growth rate and maximum cell density, cells were grown in 1 L flasks at 120 rpm of agitation and at 50 μE m^−2^ s^−1^. Specific growth rates at stationary phase are an average from three independent measurements.

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
