# Peer review of "The Chloroplast of Chlamydomonas reinhardtii as a Testbed for Engineering Nitrogen Fixation into Plants"

_ijms, 2021, doi:10.3390/ijms22168806_

Round 1

Reviewer 1 Report

The authors have written a mini-review discussing the possibilities of using Chlamydomonas reinhardtii as a test-bed for expression and engineering of nitrogenase in plants. The main arguments are the possibilities to easily transform the Chlamydomonas chloroplast genome, which could then mimic the higher plant chloroplast, and that Chlamydomonas are easy to cultivate and possible to grow under different conditions that could be relevant in order to investigate how nitrogenase could be produced as a functional complex in a photosynthetic cell. While this review is interesting, and while Chlamydomonas offers many exciting possibilities to simplify such challenging projects, the authors have not included several relevant publications, while other publications are not correctly cited. Also, the choice of using NifV is not well explained. In addition, there are many errors in figures etc that should have been noticed before/while submitting.

Major comments

  1. The authors have not discussed a highly relevant publication (https://doi.org/10.1111/1751-7915.13758) in which the Fe-only nitrogenase (the main nitrogenase discussed in this review) has been expressed in yeast.
  2. Figure 1 and legend: There are many empty boxes/squares, what are they supposed to show? The stoichiometry is not correct (N2 -> NH3). Normally the FeMo-cofactor is abbreviated as FeMo-co, not Fe-Mo-co. NifU/S should be separated as NifU and NifS (also throughout the text). The NifEN protein is pictured as a heterodimer, not a tetramer. What is the Fe-Mo-co protein, presumably the authors mean the MoFe protein? The “co” in FeMo-co stands for “cofactor”, therefore no need to say FeMo-co clusters (FeMo-cofactor clusters). Same goes for NifB-co cluster.
  3. A discussion about NifH targeted to the tobacco chloroplast showing that NifU and NifS are required is not included in this review, which is highly relevant for this discussion (doi.org/10.1111/pbi.1334).
  4. Lines 153-156: Why is NifV chosen as Nif protein to be tested in this study? Is NifV oxygen sensitive?
  5. Figure 2 and legend: Again empty boxes. Is that stoichiometry correct for the Fe-only nitrogenase (ATP)?
  6. Lines 256-257: Has to be tested experimentally.
  7. Line 269: The low activity reported in reference 25 could indicate that NifU and NifS is required.
  8. Line 273: No Supplementary Methods are reported.
  9. Line 287: Sequence similarity and structure are not necessary the same.
  10. Lines 299-301: A 3% difference does not necessarily mean that. It probably depends more on how exposed the cluster is and therefore its oxygen sensitivity. Also, interactions does not have to be the same.
  11. Lines 302-304: All relevant publications reporting about Nif expression and functionality should be mentioned, especially the isolation of functional NifH from the chloroplast (10.1038/s42003-020-01536-6, 10.1111/pbi.13347, 10.1038/s42003-020-01536-6, 10.1038/ncomms11426, etc). Reference 19 only reports expression, which is not as relevant.
  12. Figure 4: Non-cropped immunoblots should be shown. There appear to be a similar band in the TN-E lane (4C). It is surprising to see so weak signals as the HA-antibodies are usually very specific and strong, so expression must be very low. The samples for the negative control strain (TN-E) should be loaded on the same gel/membrane (4D). Also here there seems to be a similar unspecific band. In addition, the control immunoblot image seems stretched to mask the upper band?
  13. Lines 383-385: The most difficult part is likely to analyze functionality of tested Nif protein.
  14. Conclusions: Could the authors discuss the limitations of using Chlammy? For example, Fe uptake of so could perhaps be difficult to mimic?

Minor comments

  1. Overall: Genes and latin names should be in italics.
  2. Abstract, line 18: “The chloroplast of plant cells represents an attractive location for nitrogenase expression”. Why? The authors just stated in the previous sentence that nitrogenase is oxygen sensitive.
  3. Lines 96-97: The authors should include the references for these studies.
  4. Line 101: Reference 19 shows nothing experimental in this direction.
  5. Lines 101-104: The authors could mention the original publication reporting about degradation of NifD in mitochondria (10.1021/acssynbio.6b00371).
  6. Line 110: Reference 25 showed extremely low (if any) activity and only under low oxygen growth conditions.
  7. Lines 117-120: The following study should be mentioned (doi.org/10.1111/pbi1334)
  8. Line 161: synthesis
  9. Lines 161-163: assumption
  10. Line 164: Reference 21 has nothing to do with the chloroplast.
  11. Line 243: What metal cofactor is this?
  12. Lines 247-249. Partial complementation
  13. Lines 263-264: What about the 9th sulfur?
  14. Lines 313-314: Is that NifB soluble? Several reports highlight that it is difficult to accumulate soluble NifB proteins in heterologous hosts.
  15. Line 315: What does “accumulation testable” mean?
  16. Line 379: Why would anoxic induction of nifV improve yield? Is NifV degraded in the presence of oxygen?

Reviewer 2 Report

The authors discuss in this review the advantages and limitations of using Chlamydomonas reinhardtii as a photosynthetic eucaryotic model to express a functional nitrogenase enzyme using a minimal set of genes that can be used as a blueprint for engineering nitrogen fixation in plants.

The review is a valuable addition to the literature in this field. The authors have presented a clearly written manuscript, which can be accepted for publication with few revisions.

1) In the introduction mention the third strategy in to improve BNF: modify bacteria already naturally associated with cereals to improve their colonization ability, N2-fixing capabilities, and the release of NH3 produced to plant cells.

2) Explain in the text why choosing the Fe-only nitrogenase structural components.

3) C. reinhardtii, E. coli, Klebsiella oxytoca should be in italic. Please fix it when it is needed (pages 5, 6).

4) Gene names should be in italic. Please fix it when it is needed (pages 5, 6).

5) Figure 2: Add nif and vnf for the genes not only the letters H, D, G, K, …
